# IMPROVING BATCH NORMALIZATION WITH SKEWNESS REDUCTION FOR DEEP NEURAL NETWORKS

## ABSTRACT

Batch Normalization (BN) is a well-known technique used in training deep neural networks. The main idea behind batch normalization is to normalize the features of the layers (*i.e.*, transforming them to have a mean equal to zero and a variance equal to one). Such a procedure encourages the optimization landscape of the loss function to be smoother, and improve the learning of the networks for both speed and performance. In this paper, we demonstrate that the performance of the network can be improved, if the distributions of the features of the output in the same layer are similar. As normalizing based on mean and variance does not necessarily make the features to have the same distribution, we propose a new normalization scheme: Batch Normalization with Skewness Reduction (BNSR). Comparing with other normalization approaches, BNSR transforms not just only the mean and variance, but also the skewness of the data. By tackling this property of a distribution, we are able to make the output distributions of the layers to be further similar. The nonlinearity of BNSR may further improve the expressiveness of the underlying network. Comparisons with other normalization schemes are tested on the CIFAR-100 and ImageNet datasets. Experimental results show that the proposed approach can outperform other state-of-the-arts that are not equipped with BNSR.

## 1 INTRODUCTION

In recent years, deep neural networks have been applied to many visual computing tasks, such as image recognition (Krizhevsky et al., 2012; Huang et al., 2017), image super-resolution (Tong et al., 2017), video-based activity recognition (Feichtenhofer et al., 2016), etc.(Ronneberger et al., 2015; Feichtenhofer et al., 2018), achieving promising results. These models are usually trained with stochastic gradient descent or its variants. State-of-the-art neural networks often have many layers, which means they have a lot of parameters to learn, leading to practical issues including long training time and high risk of overfitting. To facilitate learning with gradient descent, Batch Normalization (BN) was proposed in (Ioffe & Szegedy, 2015), which has been found very effective in deep learning. A BN layer normalizes the batch input to zero mean and unit variance. (In practice, a BN layer learns a mapping that does not necessarily maintain the "zero mean, unit variance" property for the outputs. But that level of detail will not affect the validity of the discussion here.) This has been shown to improve the speed of convergence in training deep neural networks as well as improving the performance (He et al., 2016), and hence BN has become one common component of many popular deep networks.

We have discovered that, making the distributions of the features in the same layer more similar would make the network performs better. However, the standard BN procedure only normalizes the features to ensure that they have the same mean and variance. This does not necessarily make the distributions of the features in the same layer to become similar. For example, an exponential distribution can also have zero mean and unit variance. In other words, the standard BN, while performing normalization with respect to the mean and the variance, will not ensure the features of different layers to have similar distributions. Note that, the mean and the variance are only the first-order and second-order moments, respectively, for a distribution. To further encourage the distributions to become closer, we propose to introduce an extra dimension of normalization by mapping the data to ensure they have similar skewness. Skewness is a measure of the asymmetry of a

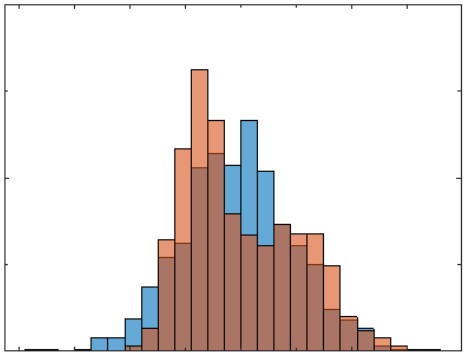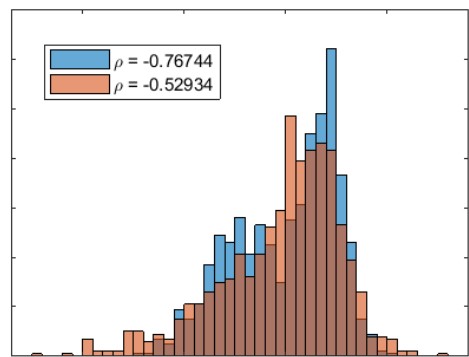

Figure 1: (Left) The figure shows two probability distributions, both with zero means and unit variances. Although they have the same mean and variance, it is obvious that they are not similar; (Right) An illustration of how $\varphi_p$ reduce the skewness. The original data X (blue) is mapped to the new data Y (orange), and the skewness is reduced.

distribution, and we hypothesize that including this measure will provide a much stronger constraint towards making these distributions become similar.

From another point of view, modifying skewness requires nonlinear operations. Recent research (Pascanu et al., 2013b; Montufar et al., 2014) has shown that deep neural networks are more expressive while stacking up the nonlinear activation with more layers. The nonlinearity introduced for modifying skewness may further contribute to improving the network's capacity in approaching any desired input-output mapping (which is typically highly nonlinear), and thus making network learning more flexible.

In this paper, we present a novel approach for improving BN with skewness reduction (BNSR) for training deep neural networks. We notice that, during training, our approach can make the feature distributions to be similar with fewer epochs. Also, we demonstrate that, it is more effective while applying BNSR on the layer with more dissimilar distributions of the features. We further compare our proposed method with other normalization schemes, including Batch Normlization (BN), Layer Normlization (LN) and Instance Normlization (IN) on CIFAR-100 and ImageNet datasets. Experimental results show that BNSR outperforms all of them. Our contributions are summarized as follows:

- We propose a new batch normalization scheme. To our best knowledge, this is the first work to consider skewness for normalization.

- The scheme introduces a nonlinear function, which not only decreases the skewness of the feature distributions, but also increases the flexibility of the network.

- We demonstrate that our approach outperforms other normalization approach on visual recognition tasks.

## 2  RELATED WORK

In this section, we first give a review for the related works on normalization, followed by a brief description of the recent understanding to BN.

### 2.1  NORMALIZATION

To shorten the training stage, researchers normalize the input data (LeCun et al., 2012). Alternatively, some initialization methods (LeCun et al., 2012; Glorot & Bengio, 2010; Wiesler & Ney, 2011) are proposed. However, such methods have their limitations as they were proposed based on strong assumptions of the feature distributions.

---

**Algorithm 1:** Training stage of BNSR, applied to features $x$ over a mini-batch

---

**Input**  : Values of $x$ over a mini-batch: $\mathcal{B} = \{x_{1...m}\}$;
**Parameters:** Parameters to be learned: $\gamma, \beta$
**Output**  : $y_i = \text{BN}_{\gamma,\beta}(x_i)$

1 $\mu_{\mathcal{B}} \leftarrow \frac{1}{m} \sum_{i=1}^{m} x_i$
2 $\sigma_{\mathcal{B}}^2 \leftarrow \frac{1}{m} \sum_{i=1}^{m} (x_i - \mu_{\mathcal{B}})^2$
3 $\hat{x}_i \leftarrow \varphi_p(\hat{x}_i)$
4 $y_i \leftarrow \gamma \hat{x}_i + \beta \equiv \text{BNSR}_{\gamma,\beta}(x_i)$

---

**Algorithm 2:** Testing stage of BNSR, applied to features $x$ over a mini-batch

---

**Input**  : Values of $x$ over a mini-batch: $\mathcal{B} = \{x_{1...m}\}$;
**Output**  : $y_i = \text{BN}_{\gamma,\beta}(x_i)$

1 Calculate the population $\mu$, $\sigma$ by unbias estimation or exponential moving average
2 **for** $i = 1 \ldots m$ **do**
3 $\quad$ $\hat{x}_i \leftarrow \frac{x_i - \mu}{\sqrt{\sigma^2 + \epsilon}}$
4 $\quad$ $\hat{x}_i \leftarrow \varphi_p(\hat{x}_i)$
5 **end**
6 $y_i = \gamma \hat{x}_i + \beta$

---

Before the proposal of BN, normalization layers like Local Response Normalization (LRN) (Lyu & Simoncelli, 2008; Jarrett et al., 2009; Krizhevsky et al., 2012), which computes the statistics of the local neighborhood for each pixel, was usually used in deep neural networks (Krizhevsky et al., 2012). Unlike LRN, Batch Normalization (Ioffe & Szegedy, 2015) normalizes the features along the batch axis, which makes the normalization more global. Besides, BN also allows higher learning rates. While large learning rates increase the scale of the weights of the network, back propagation with BN is unaffected by the scale of its weights. Also, for a given training sample, a neural network with BN does not always provide a fixed value, making BN serve as a regularization of the network. Another modification that BN made is the introduction of a pair of parameters $\beta$ and $\gamma$, which shift and scale the normalized features. These processes allow the BN layer to represent identity mapping, and increase the flexibility of the network. Since placing a ReLU layer after the BN is common, to avoid half of the neurons to be mapped to zero, it is also beneficial to do this transformation between the normalization and activation.

Meanwhile, it is worth mentioning that, in the inference stage, BN does not exactly normalize the input: the mean and variance it used are pre-computed from the training set, such that if only one sample is sent to the network (the mini-batch contains only one sample, and thus the mean is the feature itself and variance equals to zero), the BN layers can still functioning correctly.

Many normalization methods (Ba et al., 2016; Ulyanov et al., 2016; Salimans & Kingma, 2016; Luo et al., 2018; Wu & He, 2018) were proposed after BN. Layer Normalization (LN) (Ba et al., 2016) performs the normalization over all the hidden units in the same layer; Instance Normalization (IN) (Ulyanov et al., 2016) proposes to normalize each sample; Weight Normalization (WN) (Salimans & Kingma, 2016), instead of normalizing the input of the layers, operates the normalization on the filter weights; Group Normalization (GN) (Wu & He, 2018) divides channels into groups, and computes the normalization statistics of the features within each group. Comparing to BN, although these methods have their strength, they in general do not outperform BN in many visual classification/recognition problems. In addition, ELU (Clevert et al., 2015), PoLU (Li et al., 2018) and SELU (Klambauer et al., 2017) were proposed as new nonlinear activation functions, which have negative saturation $< 1$, and use this property to push the mean of the output closer to zero. Computing these activations are with lower computational complexity, and can be served as an alternative to the feature normalization.

Table 1: Comparison of error rates (%) of BNSR, BN, BN with noisy mean and variance, BN with noisy skewness on CIFAR-100. The training loss and error rate curves are in Fig. 2

|  | BNSR | BN | Noise($\mu, \sigma$) | Noise($\rho$) |
|---|---|---|---|---|
| error | 30.61 | 31.35 | 33.52 | 32.1 |

Figure 2: Comparison of performance among (1) BNSR; (2) BN; (3) BN with noisy mean and variance; (3) BN with noisy skewness on CIFAR-100. We show (a) the training loss; (b) the testing error v.s. numbers of training epochs. The model is VGG-19.

## 2.2 RECENT RESEARCH ON BN

It has been shown that (LeCun et al., 2012; Wiesler & Ney, 2011), in the training stage, if the features are whitened (the inputs have zero means, unit variances, and decorrelated) the training process can be speeded up. However, adding the whitening process for each layer is costly. To overcome this, Batch Normalization, which linear-transforms the features to have zero means and unit variances, was proposed to normalize the features.

It was believed that, the benefits of doing BN supposedly come from the reduction of the internal covariate shift effect (Ioffe & Szegedy, 2015), which is defined as the change in the distribution of the features of the layers (Ioffe & Szegedy, 2015), due to the variation of the network parameters during learning. By reducing the internal covariate shift, training can be improved. However, a recent paper (Santurkar et al., 2018) demonstrated that, the internal covariate shift effect has little to do with the effectiveness of BN. They further point out that, by both experiments and theoretical analysis, the success of BN comes from smoothness of the loss surface.

In this paper, we demonstrate that the distributions of the features in the same layer affect the performance of the network. The network is improved while these distributions are more similar. Nevertheless, as illustrated in Fig. 1a, we cannot conclude that two distributions are similar if they have the same mean and variance. Therefore, we would like to develop a new normalization scheme such that the distributions have a higher chance to become closer after normalization.

For our proposed approach, Batch Normalization with Skewness Reduction (BNSR), we adopt the advantages of original BN mentioned above. First, the features are normalized to the same mean and variance to make the loss surface smoother. Second, BNSR adopts the re-scaling parameter $\gamma$ and the re-centering parameter $\beta$ such that the network has greater flexibility. We further impose a novel step - Skewness Reduction into normalization, to encourage the distributions of the features in the same layer to become further closer. More details are to be presented in the next section.

## 3 IMPROVING BATCH NORMALIZATION WITH SKEWNESS REDUCTION

In this section, we first review the core transformations of the original BN, and then introduce our proposed approach. The basic formulation of feature normalization is based on the the following computation:

$$\hat{x} = \frac{x - \mu}{\sigma} \tag{1}$$

Table 2: Comparison of error rates (%) of BNSR, BN, LN, IN on CIFAR-100. The training loss and error rate curves are in Fig. 3

|  | BNSR | BN | LN | IN |
|---|---|---|---|---|
| error | 23.49 | 25.51 | 39.78 | 28.72 |

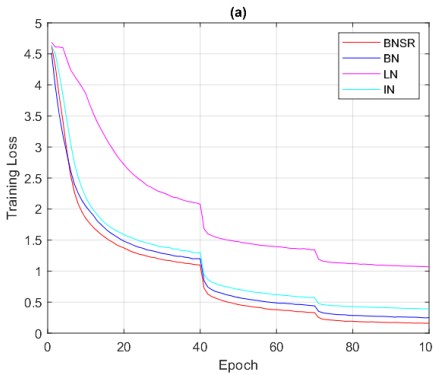 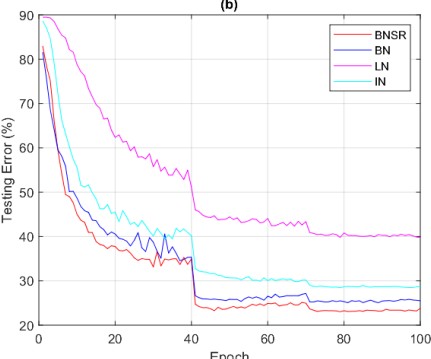

Figure 3: Comparison of performance among (1) Batch Normalization with Skewness Reduction (BNSR); (2) Batch Normalization (BN); (3) Layer Normalization (LN); (4) Instance Normalization (IN); on CIFAR-100. We show (a) the training loss; (b) the testing error v.s. numbers of training epochs. The model is ResNet-50.

where $x$ is the feature. $\mu$ and $\sigma$ in are the mean and standard deviation, which can be computed by:

$$\mu = \frac{1}{m} \sum_i x_i \tag{2}$$

$$\sigma = \sqrt{\frac{1}{m} \sum_i (x_i - \mu)^2 + \epsilon} \tag{3}$$

where $x_i$ is the $i$th element of $x$, $\epsilon$ is a small constant. The second transformation involved in BN is the scaling and shifting.

$$y = \gamma \hat{x} + \beta \tag{4}$$

where the $\gamma$ and $\beta$ are the re-scaling and re-centering parameters respectively, both being learnable.

To encourage the distributions of the features to be further similar, we propose BNSR, which adds a nonlinear function between the two parts of original BN: the feature normalization in Eq.1 and the scaling and shifting part Eq.4. We first start by giving the definitions.

**Definition 1.** *The skewness of a random variable $X$ can be defined as:*

$$\rho = \frac{3(mean - median)}{std} \tag{5}$$

The above definition is also known as the Pearson's second skewness coefficient (median skewness). In this paper, the skewness of a distribution means the skewness of the random variable that generates this distribution. Also, the concept "Skewness Reduction" points to the decrease of $|\rho|$, not $\rho$ itself. The target we want to achieve is to encourage all the $\rho$ to have a small magnitude.

For a distribution with negative skewness, which is also said to be left-skewed, the left tail is longer, and the mass of the distribution is more concentrated on the right. In contrast, a distribution with positive skewness has its mass concentrated on its left. There is no linear transformation that can reduce the skewness of a distribution. Therefore, we propose a nonlinear function to help reducing the skewness. The function is defined as follows:

**Definition 2.** *Let $\varphi_p : \mathbb{R} \rightarrow \mathbb{R}$ be a function, the skewness correction function are defined as follows:*

$$\varphi_p(x) = \begin{cases} x^p & \text{if } x \geq 0 \\ -(-x)^p & \text{if } x < 0 \end{cases} \tag{6}$$

*where $p > 1$.*

For a skewed random variable $X$ with zero mean and unit variance, there is a high probability that the main portion of the data lies in the interval (-1, 1). Applying $\varphi_p$ on $X$ pushes the data in (-1, 1) closer to zero, and make the distribution to be more symmetric, which leads to having less skewness.

As a result, after applying the step of feature normalization, we operate the step of skewness reduction, which can be described as:

$$\hat{x} \leftarrow \varphi_p(\hat{x}) \tag{7}$$

Although applying this function always leads to non-zero means and non-unit variances, these oscillations are still acceptably small if we choose a small $p$, and conceptually can be absorbed by the linear transformation right after this step. Another advantage of using these functions is due to the flexibility of the network. Since $\varphi_p$ is nonlinear, the complexity of functions that are computable by the neural network will be increased. Fig. **??**b illustrates how $\varphi_p$ reduce the skewness.

### 3.1 Hyperparameters

To implement BNSR, an extra hyperparameter $p$ is required to be determined. In order to make the distributions to become similar, we should choose a small $p$. Choosing a large $p$ may make the neural network suffer for two reasons. First, $\varphi_p$ is a contraction mapping when the input is smaller than 1. When $p$ is large, $\varphi_p$ is "over-contracted". For example, while $p = 2$, $\varphi_p$ maps 0.1 and 0.2 to 0.01 and 0.04 respectively. This may make two different features harder to be distinguished by the network, and leads to the degradation of performance. Second, large $p$ also makes the means and variances away from 0 and 1 respectively. Although the skewness reduction step always change the means and the variances, we want these changes to be small and can be absorbed by the re-scaling and re-centering parameters. Due to the above reasons, we want $p$ to be small. The default value of $p$ is set to 1.01.

### 3.2 Training and Testing with BNSR

In the training stage, we need to backpropagate the gradient of loss through the BNSR transformation. It has been proved in (Ioffe & Szegedy, 2015) that, every operation in the BN transformation is differentiable. Since the step of skewness reduction is also differentiable, we can ensure that the network can learn continuously during the training stage. As a result, any network employing BNSR can be trained using stochastic gradient descent, or its variants. During inference, like the traditional BN, the mean and variance can be obtained by using either unbiased estimation, or exponential moving average. To be precise, we present the algorithm in Alg. 1 and 2.

## 4 Experiments

In this section, we first analyze how the similarity of the feature distributions impact the performance of the neural network, by using VGG-19 (Simonyan & Zisserman, 2014) network to evaluate different settings of normalization on CIFAR-100 (Krizhevsky & Hinton, 2009). After that, we investigate the histogram for the features from different layers. We also use BNSR for only 33% of the total number of normalization layers (that is, for all the normalization layers, we use BNSR for 33% of them, and original BN for 66% of them), and analyze where BNSR is more effective.

We then evaluate BNSR with BN, LN, IN on CIFAR-100 (Krizhevsky & Hinton, 2009), and with BN on Tiny ImageNet (Russakovsky et al., 2015). All the plots and tables we present are based on five trails, we choose the median of the final accuracy. Compared with other normalization schemes, BNSR present the best performance on these two datasets. All experiments are implemented using Pytorch 1.0.1 (Paszke et al., 2017) with Python 3.6, on a machine with Ubuntu 18.04, Intel CPU E5-2603, and a single nVidia GTX 1080 GPU with cuda 9.0.

### 4.1 CIFAR-100 dataset

The CIFAR-100 dataset contains 60,000 color images with size $32 \times 32$, which contains 50,000 training and 10,000 testing samples. We use ResNet-50 to evaluate our proposed approach on this

Table 3: Comparison of error rates (%) of BN and BNSR on ImageNet dataset. The training loss and error rate curves are in Fig. 4

|  | BNSR | BN |
|---|---|---|
| error | 38.32 | 39.54 |

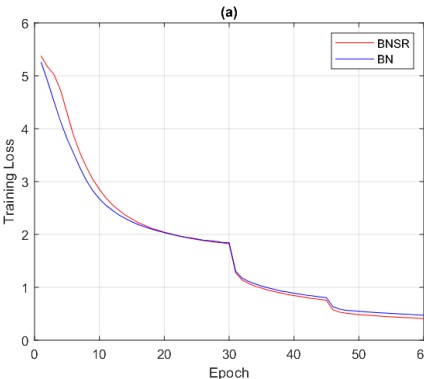 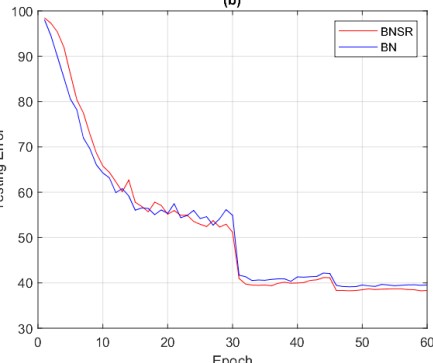

Figure 4: Comparison of performance between BNSR and BN on Tiny ImageNet dataset. We show (a) the training loss; (b) the testing error v.s. numbers of training epochs. The model is ResNet-50.

dataset. The network is trained for 100 epochs using stochastic gradient descent, with momentum equals to 0.9, and batch size equals to 50. The initial learning rate is set to 0.1, and decayed by a factor of 10 at the $41th$ and $71th$ epoch. During training, the images are first cropped with size equals to $32 \times 32$ at random location with padding equals to 4, followed by a random horizontal flip.

**Impact of the similarity of the feature distributions:** We hypothesize that, the more similar the distributions of the features in the same layer, the better the network. So, we analyze the importance of the similarity of the distributions which are in the same layer. We insert different mappings into the BN layers (right before the re-centering and re-scaling mapping), by using the following settings:

- $x \leftarrow x$ (identity mapping)
- $x \leftarrow ax + b$ where $a, b \sim N_m(0, 0.5)$
- $x \leftarrow \varphi_p(x)$ where $p \sim Unif_m(1, 1.05)$
- $x \leftarrow \varphi_p(x)$ where $p = 1.01$

The first and the forth setting represent the conventional BN and BNSR respectively, while for the second and third setting add some noise to the features such that they follow dissimilar distributions. To be precise, for a layer with $m$ channels of output, $a$, $b$ and $p$ are $m$-dimensional vectors, following the normal distribution and uniform distribution respectively.

We analyze the performance of VGG-19 (Simonyan & Zisserman, 2014) on CIFAR-100. Fig. 2 and Tab. 1 show the training loss/error curves, and the final results respectively. The results suggest that, the network performs better if their distributions of the features in the same layer are more similar.

**Comparison of other normalization methods:** We experiment BNSR with other feature normalization methods, including BN, LN and IN, on a ResNet-50 model. Fig. 3 shows the training loss and the error curves, and Tab. 2 shows the final results. We can see that BNSR outperforms original BN by about 2%. This is an encouraging result, as for the recent research on activation function, the improvement for using the same network with an improved activation function is still $< 1\%$ (Ramachandran et al., 2018).

**Features in the earlier layers:** If our hypothesis of the impact of the similarity of the feature distributions is true, applying BNSR may not be that useful to the layer which already has similar feature distributions. For this reason, we investigate the distributions of the features (the output of Eq. 1). Features from different layers are collected. We discovered that, after some epochs of training using BN, the distributions from the later layers become similar with a faster speed,

comparing to these from the earlier layers. We conjecture that encouraging the features which have more dissimilar distributions (the ones in earlier layers) to be similar may leads to greater improvement of the quality of learning, and we experiment on ResNet-50 by using BNSR on only 1/3 of the normalization layers, by three different settings:

- BNSR is used for all layers uniformly;

- BNSR is used only for the earlier layers;

- BNSR is used only for the later layers;

The settings with BNSR located at the earlier layers has gained greater improvement, which suggests our hypothesis is rational. We can also observe that after adding skewness reduction step, the distribution become similar in a faster rate. The results of the experiments are provided in the appendix.

### 4.2 TINY IMAGENET DATASET

We experiment our BNSR in the Tiny ImageNet dataset, which is a subset of ImageNet classification dataset (Russakovsky et al., 2015). The original dataset contains more than 1.2 million training samples belonging to 1000 classes, while the Tiny dataset has 200 classes, each class contains 500 training images, 50 validation images, and 50 test images. We test on the $50 \times 200 = 10000$ validation images, using the ResNet-50 model (He et al., 2016). The network is trained for 60 epochs using stochastic gradient descent, with momentum equals to 0.9, and batch size equals to 50. The initial learning rate is 0.1, and decayed by a factor of 10 for $31th$, $46th$ epoch. In the training stage, the images are first normalized, and are cropped with size equals to $64 \times 64$ with padding equals to 4, followed by a random horizontal flip with probability equals to 0.5. For the testing images, only normalization are performed. Fig. 4 shows the learning situation of BN v.s. BNSR, The testing error is can be found in Tab. 3.

## 5 TIME COMPLEXITY OF BNSR

Due to the computation for the Skewness Reduction steps, the time used for training a network with BNSR is greater than the one with regular BN. In terms of wall clock time, BNSR requires 113s v.s. BN with 86s for 1 epoch on CIFAR-100 using ResNet-50. However, the difference is still not significant for inference as the time complexity for the extra step is equals to $O(n)$. Also, in the previous section we have already discussed that, using fewer numbers of BNSR layers for the normalization has already provided a great improvement for the accuracy. So we can use fewer BNSR layers to shorten the training time.

## 6 CONCLUSION AND FUTURE WORKS

We proposed an normalization scheme - Batch Normalization with Skewness Reduction (BNSR) - for faster and improved learning in deep neural networks. Besides adopting the advantages from using regular BN, BNSR uses a nonlinear function to modify the skewness after the features are normalized to have zero means and unit variances. Different from other normalization approaches, like Layer Normalization (LN), Weight Normalization (WN), Instance Normalization (IN), in BNSR attention is still on how to normalize the data in the batch dimension. Comparing to traditional BN, BNSR considers not just the mean and variance, but also the skewness. This was motivated by the observation that, two distributions having their mean and variance equal does not imply they are similar. By reducing the skewness, the features are encouraged to have more similar distributions. Also, as the function for reducing the skewness is nonlinear, applying it on the feature also make the network to become more expressive. Experimental results also show that BNSR outperforms other state-of-the-art normalization approaches,

We have not explored all the possibilities of BNSR. Our future work includes applying the Skewness Reduction concept to the normalization in Recurrent Neural Networks (RNN) (Pascanu et al., 2013a), as the internal covariate shift may serve differently from traditional CNN.

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

## A  APPENDIX

We provide the experimental result mentioned in Sec. 4: "Features in the earlier layers" here. Fig. 6 shows the distributions of the features from both earlier and later layers. Fig. 5 shows the plots of training loss and testing error, and Tab. 4 presents the final error. Fig. 7 shows the histogram where BNSR is used.

Table 4: Comparison of error rates (%) of BNSR under different percentage of usage on CIFAR-100. The training loss and testing error plots can be found in Fig. 5.

|  | 100% | 33%(uni) | 33%(early) | 33%(late) |
|---|---|---|---|---|
| error | 23.49 | 23.40 | 23.74 | 25.20 |

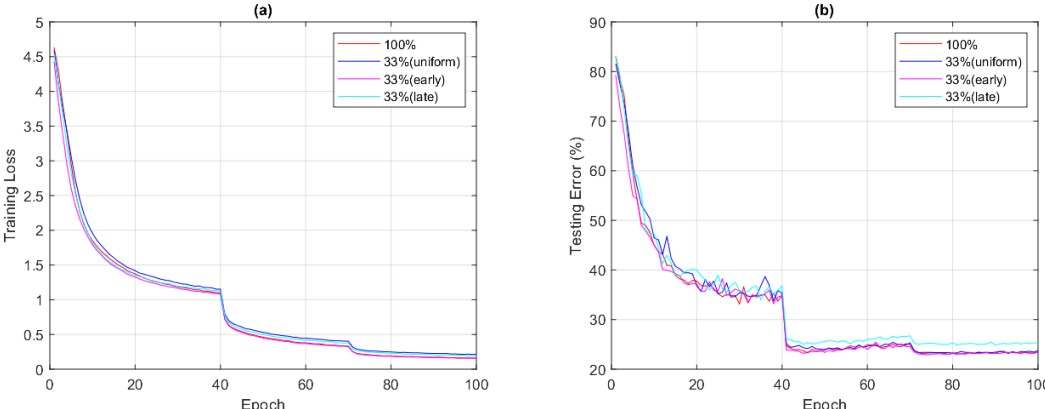

Figure 5: Comparison of performance of BNSR under different percentage of usage on CIFAR-100. (1) BNSR are used for all the normalization layers; (2) BNSR are used for 1/3 of the normalization layers uniformly; (3) BNSR are used for 1/3 of the normalization layers (earlier layers); (4) BNSR are used for 1/3 of the normalization layers (late layers). We show (a) the training loss; (b) the testing error v.s. numbers of training epochs. The model is ResNet-50.

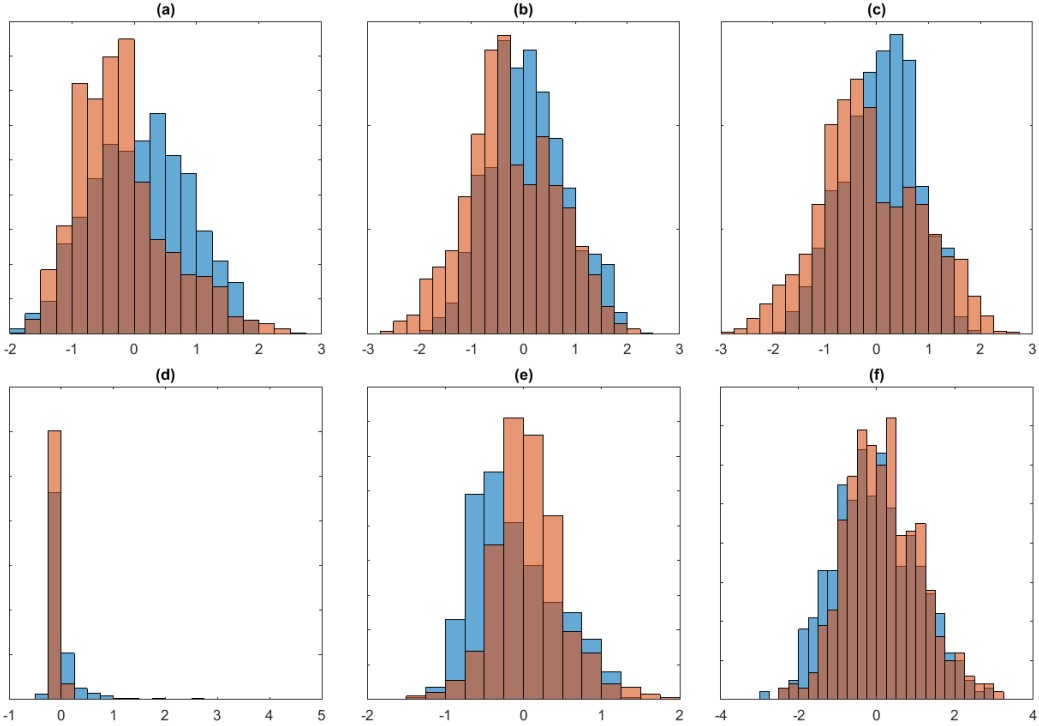

Figure 6: The histograms of the features in BN layers. (a)-(c) show the histograms of two features in the same layer (earlier part) at epoch = 1, 5, 15; (d)-(f) show the histograms of two features in the same layer (later part) at epoch = 1, 5, 15.

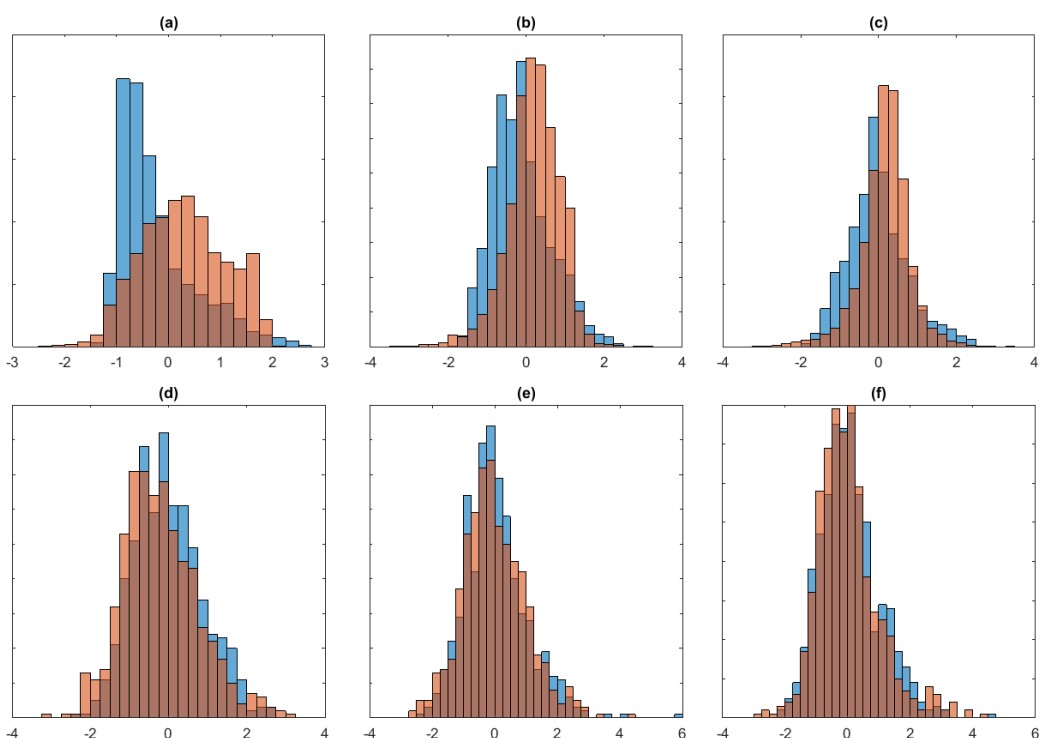

Figure 7: The histograms of the features in BNSR layers. (a)-(c) show the histograms of two features in the same layer (earlier part) at epoch = 1, 5, 15; (d)-(f) show the histograms of two features in the same layer (later part) at epoch = 1, 5, 15.

