# OpenReview forum: "Improving Batch Normalization with Skewness Reduction for Deep Neural Networks"
_ICLR.cc/2020/Conference — Reject_

### Official Review · AnonReviewer1 · 2019-10-21
**Official Blind Review #1**

**Rating:** 3

**Review:**

This paper proposed to improve the regular batch normalization by reducing the skewness of the hidden features. To this end, the authors introduce a non-linear function to reduce the skewness. However, the analysis and experiments are too weak to support the authors' claim.

1. The motivation is not clear. Why is it necessary to reduce the skewness? There is no practical or theoretical evidence to support it.

2. To verify the proposed non-linear transformation can reduce skewness, it's better to visualize the learned feature distribution to confirm this point.

3. Experiments with more datasets and networks are needed to evaluate the performance of the proposed method.

**Experience Assessment:**

I have published one or two papers in this area.

**Review Assessment: Checking Correctness Of Derivations And Theory:**

I assessed the sensibility of the derivations and theory.

**Review Assessment: Checking Correctness Of Experiments:**

I assessed the sensibility of the experiments.

**Review Assessment: Thoroughness In Paper Reading:**

I read the paper thoroughly.

---

### Official Review · AnonReviewer3 · 2019-10-23
**Official Blind Review #3**

**Rating:** 3

**Review:**

This paper develops an improved Batch Normalization method, called BNSR. BNSR applies a nonlinear mapping to modify the skewness of features, which is believed to keep the features similar, speedup training procedure, and further improve the performance.

I have several concerns:
1.	To investigate the impact of the similarity of the feature distributions, the author proposes four settings, including BN and BNSR. However, the added noise of last three settings not only makes the feature dissimilar but also breaks the nature of zero mean and unit variance. This is still unclear whether the similar distribution of features make BNSR outperform BN.
2.	The skewness is used to measure the asymmetry of the probability distribution, but not the similarity between two different distributions. Distributions with zero mean, unit variance, and near zero skewness could still be very different.
3.	Based on “for ... X with zero mean and unit variance, there is a high probability that ... lies in the interval (-1, 1)”, the paper introduces φ_p(x), where p > 1, to decrease the skewness of the feature map x. However, there are about 32% elements of X that their absolute values are larger than 1, for a standard normal distribution. Figure 6 & 7 also show that for real features in neural network, there are a significant number of elements that lie out of (-1, 1). Will this lead to instability during training? To better understand the effect of φ_p(x), I think ρ (the Pearson’s second skewness coefficient), right before and after φ_p(x), should be shown for each layer at several epochs.
4.	The results on CIFAR-100 and Tiny ImageNet are not convincing enough in my opinion. Some further experiments on ImageNet with a reasonable baseline will makes the results more convincing.


**Experience Assessment:**

I have published in this field for several years.

**Review Assessment: Checking Correctness Of Derivations And Theory:**

I assessed the sensibility of the derivations and theory.

**Review Assessment: Checking Correctness Of Experiments:**

I carefully checked the experiments.

**Review Assessment: Thoroughness In Paper Reading:**

I read the paper thoroughly.

---

### Official Review · AnonReviewer2 · 2019-10-23
**Official Blind Review #2**

**Rating:** 3

**Review:**

The paper proposes to add an extra nonlinearity function in batch normalization, between the normalization and affine scaling. The nonlinearity is a power function x ** p for x >= 0 and - (-x) ** p for x < 0 (python pseudo-code), where p is a constant, and the authors propose to keep it fixed to 1.01. The intuition behind is reducing skewness of activations, and the modification is evaluated on CIFAR and tiny ImageNet datasets. The authors also experiment with the part at which to insert the nonlinearity.

I propose reject mainly due to insufficient experimental evaluation. The authors choose small datasets on which trained networks have large variance, and report a single accuracy value for each network, so it is not possible to judge the effectiveness of the method. Regarding the skewness reduction itself, it is not very convincing too, because the authors use a very small value for p (1.01), and because after affine layer ReLU removes most of the negative part, so perhaps the negative part of the nonlinearity is not needed.

I would suggest the following experiments to improve the paper:
 - try various values of p
 - try removing negative part of the nonlinearity
 - report mean+-std results of 5 runs for each experiment on small dataset
 - include experiments on full scale ImageNet
 - include more network architectures, or at least have more configurations of ResNet.

Would be very helpful if authors included code in pytorch or tensorflow for the proposed modification. It is surprising that such a small addition increases epoch time from 86 s to 119 s.

Also, there are some minor grammar mistakes. There is an undefined figure reference on page 6, and it is not clear what figures 6 and 7 are supposed to show, since the colors of histograms are never explained.

**Experience Assessment:**

I have published in this field for several years.

**Review Assessment: Checking Correctness Of Derivations And Theory:**

I assessed the sensibility of the derivations and theory.

**Review Assessment: Checking Correctness Of Experiments:**

I carefully checked the experiments.

**Review Assessment: Thoroughness In Paper Reading:**

I read the paper at least twice and used my best judgement in assessing the paper.

---

### Decision · Program_Chairs · 2019-12-19

**Decision:**

Reject

**Comment:**

The paper proposes a novel mechanism to reduce the skewness of the activations. The paper evaluates their claims on the CIFAR-10 and Tiny Imagenet dataset. The reviewers found the scale of the experiments to be too limited to support the claims. Thus we recommend the paper be improved by considering larger datasets such as the full Imagenet. The paper should also better motivate the goal of reducing skewness.